# Adaptation after vastus lateralis denervation in rats demonstrates neural regulation of joint stresses and strains

Cristiano Alessandro[1]*, Benjamin A Rellinger[2], Filipe Oliveira Barroso[1], Matthew C Tresch[1,2,3,4]*

[1]Department of Physiology, Northwestern University, Chicago, United States; [2]Department of Biomedical Engineering, Northwestern University, Evanston, United States; [3]Department of Physical Medicine and Rehabilitation, Northwestern University, Chicago, United States; [4]Shirley Ryan AbilityLab, Chicago, United States

**Abstract** In order to produce movements, muscles must act through joints. The translation from muscle force to limb movement is mediated by internal joint structures that permit movement in some directions but constrain it in others. Although muscle forces acting against constrained directions will not affect limb movements, such forces can cause excess stresses and strains in joint structures, leading to pain or injury. In this study, we hypothesized that the central nervous system (CNS) chooses muscle activations to avoid excessive joint stresses and strains. We evaluated this hypothesis by examining adaptation strategies after selective paralysis of a muscle acting at the rat's knee. We show that the CNS compromises between restoration of task performance and regulation of joint stresses and strains. These results have significant implications to our understanding of the neural control of movements, suggesting that common theories emphasizing task performance are insufficient to explain muscle activations during behaviors.
DOI: https://doi.org/10.7554/eLife.38215.001

*For correspondence:
cristiano.alessandro@northwestern.edu (CA);
m-tresch@northwestern.edu (MCT)

Competing interests: The authors declare that no competing interests exist.

## Introduction

The question of how the central nervous system (CNS) determines motor commands is usually answered in terms of task performance (*Todorov and Jordan, 2002*; *Guigon et al., 2007*; *Scott, 2004*): the CNS activates muscles according to how they contribute to task goals such as grasping an object or escaping a predator. Additional criteria based on energetics (*Todorov and Jordan, 2002*; *Izawa et al., 2008*; *Kurtzer et al., 2006*), dynamics (*Uno et al., 1989*), or kinematics (*Flash and Hogan, 1985*) can further constrain muscle activations, and the specific criteria used by the CNS to specify muscle activations underlying behavior has been the topic of considerable research (*Scott, 2004*; *Todorov, 2004*; *Alessandro, 2016*; *Kistemaker et al., 2014*; *Prilutsky, 2000*).

This previous work, however, has generally ignored another critical aspect of muscle actions: how muscles act on internal joint structures such as ligaments or articular cartilage between bones (*Herzog et al., 2003*). Although muscle actions on internal joint structures have been considered in clinical biomechanics or sports medicine in the context of injuries (*Farrokhi et al., 2013*; *Pal et al., 2012*; *Felson, 2013*; *Konrath et al., 2017*), their more general role in the context of the neural control of movement and how they are balanced with task goals remains poorly understood. We examine these issues in this study, evaluating the hypothesis that the nervous system chooses muscle activations to achieve task performance while minimizing internal joint stresses and strains.

Evaluating whether the CNS regulates stresses and strains within internal joint structures is challenging, primarily because of the complexity of joint mechanics. Since muscles can only affect task

**eLife digest** Although most of us will never achieve the grace and dexterity of professional ballerina Misty Copeland, we each make sophisticated, complex movements every day. Even simple movements often involve coordinating many muscles throughout the body. Moreover, because we have so many muscles, there are often multiple ways that we could use them to make the same movement. So which ones do we use, and why?

Many studies into muscle control focus on how the muscles activate to perform a task like kicking a soccer ball. But muscles do more than just move the limbs; they also act on joints. Contracting a muscle exerts strain on bones and the ligaments that hold joints together. If these strains become excessive, they may cause pain and injury, and over a longer time may lead to arthritis. It would therefore make sense if the nervous system factored in the need to protect joints when turning on muscles.

The quadriceps are a group of muscles that stretch along the front of the thigh bone and help to straighten the knee. To investigate whether the nervous system selects muscle activations to avoid joint injuries, Alessando et al. studied rats that had one particular quadriceps muscle paralyzed. The easiest way for the rats to adapt to this paralysis would be to increase the activation of a muscle that performs the same role as the paralyzed one, but places more stress on the knee joint. Instead, Alessando et al. found that the rats increase the activation of a muscle that minimizes the stress placed on the knee, even though this made it more difficult for the rats to recover their ability to use the leg in certain tasks.

The results presented by Alessando et al. may have important implications for physical therapy. Clinicians usually work to restore limb movements so that a task is performed in a way that is similar to how it was done before the injury. But sometimes repairing the damage can change the mechanical properties of the joint – for example, reconstructive surgery may replace a damaged ligament with a graft that has a different strength or stiffness. In those cases, performing movements in the same way as before the surgery could place abnormal stress on the joint. However, much more research is needed before recommendations can be made for how to rehabilitate rats after injury, let alone humans.

DOI: https://doi.org/10.7554/eLife.38215.002

performance by acting through joint structures (*Herzog et al., 2003*), any manipulation affecting joint structures will usually also affect task performance. For example, damage to a ligament (*Gutierrez et al., 2009*; *Needle et al., 2014*; *O'Connor et al., 1993*) will alter both the distribution of stresses and strains within the joint (*Chen et al., 2017*; *Gardinier et al., 2013*) as well as how muscle forces are transmitted across the joint (*Boeth et al., 2013*). Similarly, although removal of sensory feedback from joint structures might compromise neural control of joint stresses and strains (*O'Connor et al., 1993*; *O'Connor et al., 1992*; *Salo et al., 2002*; *Solomonow, 2006*; *Solomonow and Krogsgaard, 2001*; *Johansson et al., 1991*), this sensory feedback can also provide information about task performance (*Ferrell et al., 1985*; *Sjölander et al., 2002*). These challenges have made it difficult to reach clear conclusions as to whether the CNS controls internal joint stresses and strains.

In the experiments reported here, we overcome these challenges by taking advantage of the unique properties of the rat knee joint (*Figure 1a*). Activation of the quadriceps muscles vastus lateralis (VL) and vastus medialis (VM) in the rat produces very similar forces at the distal tibia and so will produce similar joint torques (*Sandercock et al., 2018*). These muscles therefore have redundant contributions to task performance. On the other hand, VL and VM produce opposing mediolateral forces on the patella (*Sandercock et al., 2018*; *Lin et al., 2004*; *Wilson and Sheehan, 2010*). In the rat, these mediolateral patellar forces are not transmitted to the tibia but are balanced by contact forces between the patella and the femur within the trochlear groove (*Sandercock et al., 2018*). To avoid overloading patellofemoral contact forces and prevent joint pain or injury, the CNS should therefore balance activation of VM and VL to minimize net mediolateral forces on the patella while producing the knee torque necessary to achieve task goals. Since VM and VL in the rat have redundant contributions to task performance but opposing contributions to joint stresses, we can use the

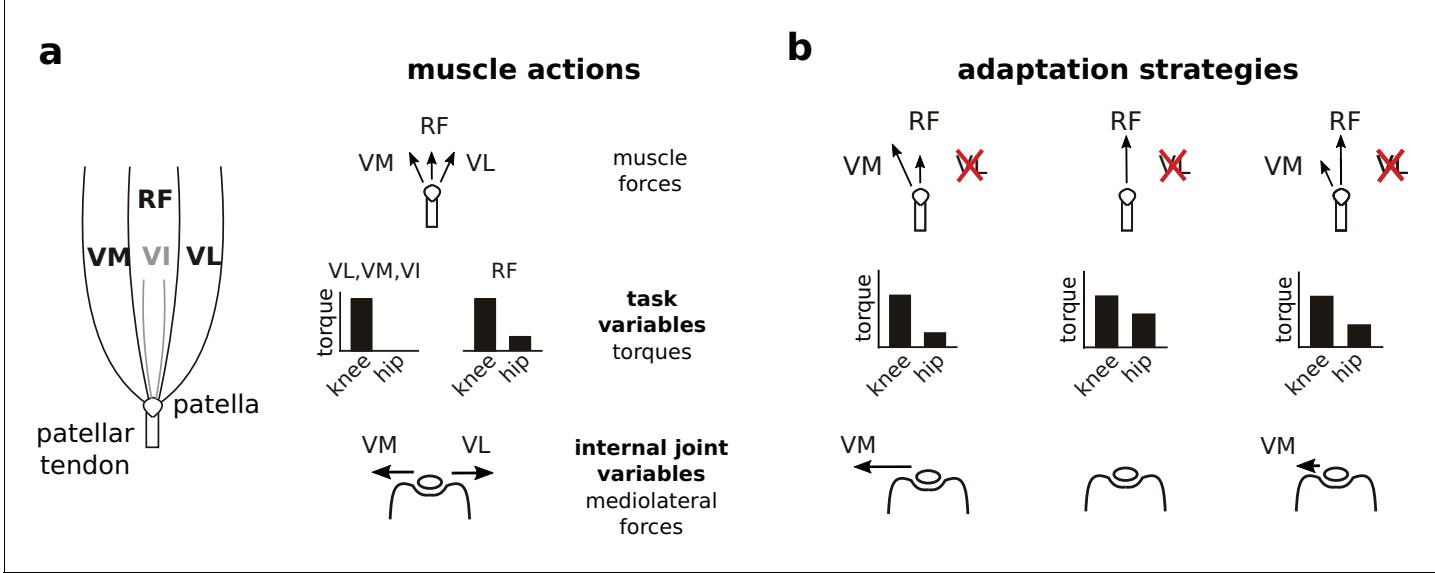

**Figure 1.** Actions of quadriceps muscles on task performance variables and internal joint state variables, and possible adaptation strategies after VL paralysis. The left schematic in (a) illustrates the anatomical organization of the quadriceps muscles; VI is indicated in gray because it is located underneath RF. The right schematics depict the force directions produced by each of these muscles on the patella (top), the corresponding joint torques (middle), and the mediolateral patellar forces produced by VM and VL (bottom). Note that RF produces torques at both the hip and knee, and produces negligible mediolateral force on the patella. VM and VL have redundant contributions to knee torque (task variables), and opposite contributions to mediolateral patellar forces (internal joint variables). Panel (b) illustrates alternate adaptation strategies for compensating to paralysis of VL. The left column shows the consequences of an adaptation strategy prioritizing regulation of task performance variables: increasing VM activation restores joint torques and therefore joint kinematics, but would create an unbalanced mediolateral force on the patella. The middle column shows the consequences of an adaptation strategy prioritizing regulation of internal joint state variables: increasing RF activation and reducing VM activation eliminates mediolateral forces on the patella, but would create aberrant joint torques at the hip. The right column shows the consequences of an adaptation strategy that considers both task performance and internal joint state, preferentially increasing RF to minimize mediolateral patellar forces while maintaining VM activation to limit deviations in joint torques.
DOI: https://doi.org/10.7554/eLife.38215.003

rat knee joint to design tractable experiments to evaluate whether the CNS regulates joint stresses and strains.

We examined neural adaptation strategies following paralysis of VL (*Figure 1b*). If the CNS prioritizes task performance, the easiest adaptation strategy would be to increase activation of VM. Because VM and VL have redundant contributions to task performance, this strategy would restore joint kinematics without requiring changes in other muscle activations. This straightforward restoration of task performance, however, would come at the cost of unbalanced mediolateral patellar forces. On the other hand, if the CNS prioritizes joint integrity, it should decrease activation of VM and increase activation of rectus femoris (RF) since RF produces minimal mediolateral force on the patella. This strategy to minimize joint stresses, however, would come at the cost of a more complex, potentially incomplete restoration of task performance: increased activation of RF will restore the knee torque lost by VL paralysis but will also introduce an additional hip flexion torque. This strategy will therefore either require compensatory activity in hip extensor muscles or introduce residual deviations in joint kinematics. Finally, if the CNS considers both task performance and internal joint stresses, it might preferentially increase RF activation while maintaining activation of VM. This strategy would minimize mediolateral forces on the patella while also limiting deviations of joint kinematics, reflecting a compromise between control of task performance and internal joint state. Observing either of the last two strategies would demonstrate that the CNS considers internal joint stresses since the preferential increase of RF activation reflects minimization of mediolateral patellar forces. Note that vastus intermedius (VI) is very small in the rat and so is unlikely to contribute substantially to the adaptation following VL paralysis. Our results show that rats compensated for the loss of VL by preferentially increasing activation of RF, consistent with the hypothesis that the CNS chooses muscle activations to minimize stresses and strains to internal joint structures.

# Results

## Paralysis of VL by peripheral nerve cut

Cutting the branch of the quadriceps nerve innervating VL abolished the large majority of VL activation during locomotion, as illustrated in *Figure 2a–b*. The loss of EMG persisted over the 7 week adaptation period, demonstrating the absence of VL reinnervation. Consistent with this loss of activation, the mass of the denervated VL measured at the end of the experiment (i.e. 8 weeks after nerve cut) was significantly lower (p<0.001) than the mass of the intact contralateral VL (*Figure 2c*), reflecting atrophy of the paralyzed muscle. These results demonstrate that our nerve cut procedures effectively paralyzed VL.

## Preferential increase in RF activation following VL paralysis

Paralysis of VL will cause a reduced ability to balance the medial forces on the patella produced by VM, resulting in aberrant mechanical stresses within the knee joint. As illustrated in *Figure 1b*, we predicted that if the CNS acts to minimize these internal joint stresses, it should compensate for VL

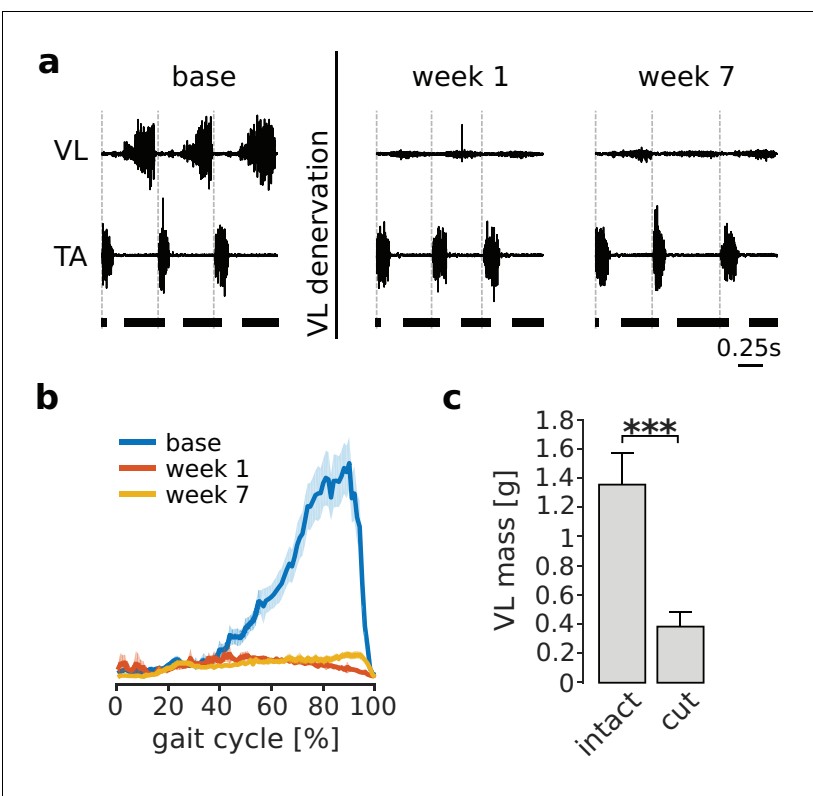

**Figure 2.** Paralysis of VL by denervation. Panel (**a**) illustrates EMGs from VL and tibialis anterior (TA) during treadmill locomotion in one rat before (base), immediately after (week 1), and several weeks after (week 7) VL denervation. The vertical dashed lines in each plot indicate the onset of TA activity, and the horizontal thick bars at the bottom indicate the stance phase. VL activity was largely abolished after nerve cut, while that of other muscles (indicated here by TA) persisted. The small residual activity in VL was likely due to motion artifact or low-level cross talk. The averaged EMG envelope of VL over many strides (95% confidence interval for the mean, $N_s$ = 104, 102, 88 for base, week 1 and week 7 respectively) confirms this result (**b**). The mass of the denervated VL, measured 8 weeks after nerve cut, was significantly lower for all animals as compared to the mass of the intact VL in the contralateral hindlimb. Bars are averages ± standard deviations (s.d.) across animals; N = 6 for each bar (**c**). ***p<0.001.

DOI: https://doi.org/10.7554/eLife.38215.004

The following source data is available for figure 2:

**Source data 1.** Mass of denervated and intact VL.

DOI: https://doi.org/10.7554/eLife.38215.005

paralysis by preferentially increasing activation of RF. An example of RF and VM activations before and after VL paralysis is shown in *Figure 3a–b* for one animal. In the first week after nerve cut, there was a large increase of the activation of both RF and VM during stance. Over subsequent weeks of adaptation, the activation of both muscles decreased. By 7 weeks after the nerve cut, the activation of VM during stance returned to levels similar to those observed in baseline conditions. At this same time point, however, the activation of RF remained elevated as compared to baseline conditions, consistent with our predictions based on the neural control of internal joint stresses.

This preferential increase in RF activation during stance was consistent across animals, as illustrated in *Figure 3c*. While the activity of VM was not significantly different from baseline at any time after VL nerve cut ($p_{week1}$ = 1, $p_{week2}$ = 1, $p_{week7}$ = 0.154), RF activity was significantly increased at all time points ($p_{week1}$ = 0.007, $p_{week2}$ < 0.001, $p_{week7}$ < 0.001). More importantly, the change in RF activation was significantly larger than the change in VM activation at all time points ($p_{week1}$ < 0.001, $p_{week2}$ < 0.001, $p_{week7}$ < 0.001). These results demonstrate that the CNS preferentially increased activation of RF following VL paralysis, consistent with the neural regulation of internal joint stresses.

## Selective increase in RF muscle mass following VL paralysis

The persistent increased activation of RF that we observed following nerve cut should induce use-dependent muscle growth. To evaluate this possibility, we measured quadriceps muscle masses 8 weeks after VL paralysis (*Figure 4*). We found that there was a significant increase in RF muscle mass in the affected limb as compared to the mass of RF in the unaffected limb (p=0.047). In contrast, the masses of VM and VI were not significantly different between the two limbs (VM: p=1; VI: p=1).

## Adaptation in overall joint kinematics following VL paralysis

VL paralysis caused initial changes in overall joint kinematics that were compensated for over the subsequent weeks of adaptation. *Figure 5b* shows the average joint angle trajectories for one animal before and after VL paralysis. There was an initial reduction in the stance phase duration and in the range of motion at the knee, consistent with the reduction of knee extensor torque during stance caused by VL paralysis. The range of motion at the ankle, on the other hand, increased. In this animal, the range of motion at the hip was minimally affected. At later time points, all of these measures of overall joint kinematics recovered to levels similar to those observed before VL paralysis, although there were persistent differences in the specific joint angle trajectories (see below).

The changes in these measures of overall joint kinematics, averaged across all animals, are shown in *Figure 5c*. The duration of the stance phase, expressed as a percentage of the entire gait cycle, was significantly reduced in the first week after VL paralysis ($p_{week1}$ <0.001), but at later time points returned to levels that were not significantly different from baseline ($p_{week2}$ = 0.457; $p_{week7}$ = 0.192). The ranges of motion of the hip and knee were significantly smaller in the first week after VL paralysis (hip: $p_{week1}$ = 0.008; knee: $p_{week1}$ <0.001), while the range of motion at the ankle was significantly larger ($p_{week1}$ <0.001). At week 2, the hip and the knee ROM remained significantly lower than baseline (hip: $p_{week2}$ = 0.007; knee: $p_{week2}$ <0.001) whereas the ankle was no longer significantly increased ($p_{week2}$ = 0.059). By week 7, the ROM of the hip and ankle joints were not significantly different from baseline (hip: $p_{week7}$ = 0.502; ankle: $p_{week7}$ = 0.781) but there was a modest, though significant, decrease in the knee range of motion ($p_{week7}$ = 0.009). These results show that overall joint kinematics are initially impaired following VL paralysis. Over the subsequent weeks of adaptation, these deficits largely disappear and overall locomotor function is restored to baseline levels, although there is a persistent deficit in knee function.

## Persistent changes in joint kinematics following VL paralysis

The preferential increase in RF activation shown in *Figure 3* introduces an extra flexion torque at the hip. If the CNS preserves individual joint angles, this extra torque should be compensated by increased activation of hip extensor muscles (e.g. BFp, SM, and GRc). However, we found no evidence for increased hip extensor muscle activation at any point of the adaptation (*Figure 3—figure supplement 1*), suggesting that the CNS does not preserve individual joint angle trajectories. To evaluate this possibility more directly, we examined the detailed joint kinematics through the adaptation period.

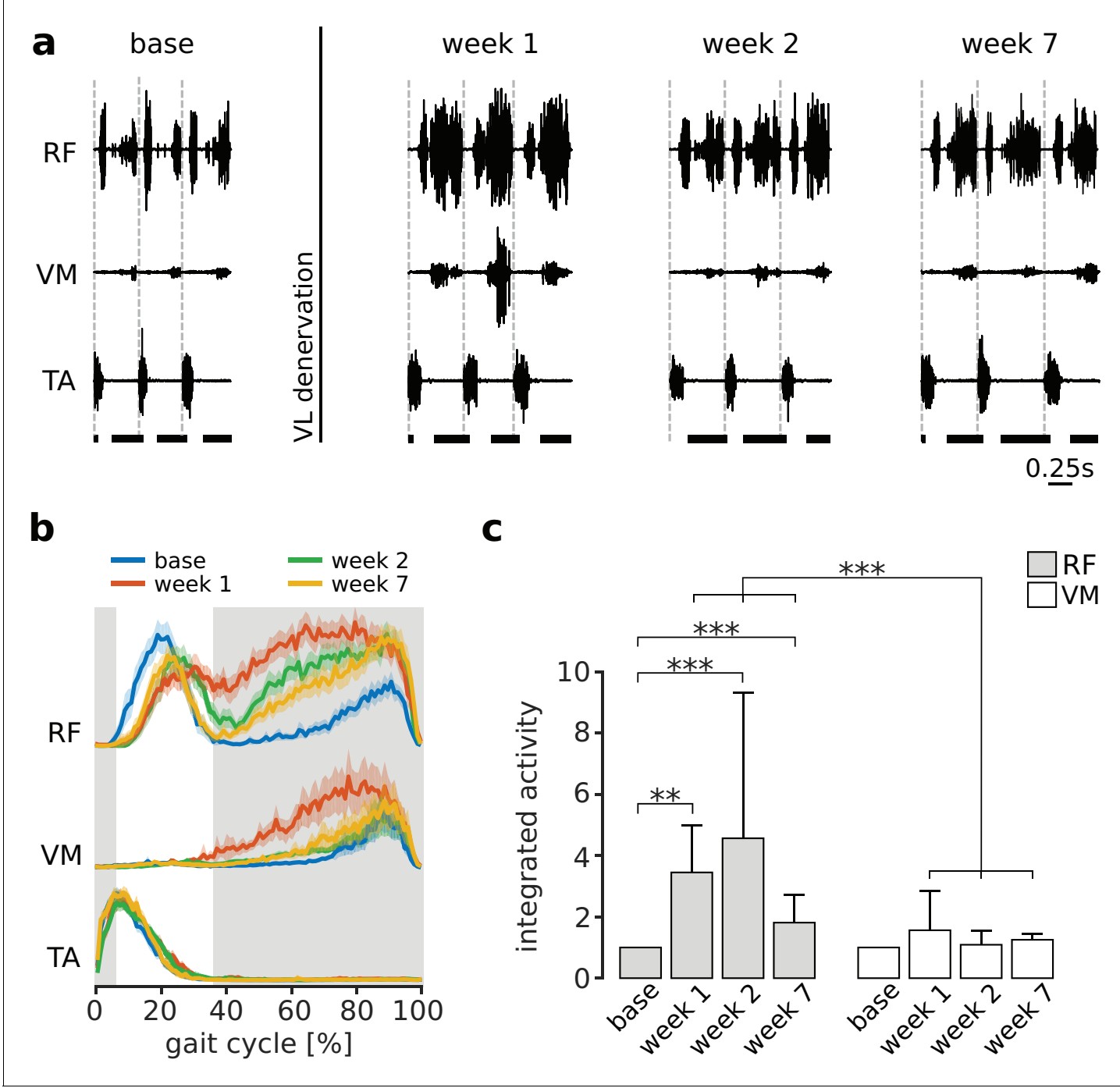

**Figure 3.** Preferential increase of RF activation following VL paralysis. Panel (**a**) illustrates locomotor activity in RF and VM before (base) and after (week 1, week 2 and week 7) VL denervation for one animal. Activation of TA is shown to indicate the onset of each gait cycle (dashed vertical lines), and the horizontal thick bars at the bottom indicate the stance phase of locomotion. In the first week after VL denervation, activity of both RF and VM increased. At later weeks, activity in VM reduced to levels similar to baseline, whereas activation of RF remained elevated. Panel (**b**) illustrates the EMG envelopes of RF, VM and TA in this same animal (95% confidence interval for the mean activity, $N_s$ = 104, 102, 96, 88 strides for base, week 1, week 2, and week 7 respectively), and confirms the results shown in (**a**). The shaded region indicates the stance phase for the baseline condition. Panel (**c**) reports the integrated activity during stance of RF and VM before (base) and after VL paralysis (week 1, 2, 7) for all animals (averages ± s.d. across animals), expressed as a multiple of the integrated activity observed in baseline conditions. Note that RF activation was larger than VM activation at each week after VL paralysis. For RF: N = 5, 4, 5, and 3 animals contributed to the bars at base, week 1, week 2, and week 7, with an average number of strides of $N_s$ = 65, 94, 64 and 99 respectively. Note that because of exclusion criteria (see Materials and methods), not every animal contributed data to *Figure 3 continued on next page*

*Figure 3 continued*

each bar. For VM: N = 5, 4, 5, and 4 animals, and $N_s$ = 74, 98, 53, 93 strides contributed to the same time points. Significance levels: **p<0.01; ***p<0.001.

DOI: https://doi.org/10.7554/eLife.38215.006

The following source data and figure supplements are available for figure 3:

**Source data 1.** Integrated activity of quadriceps muscles during stance phase of locomotion.

DOI: https://doi.org/10.7554/eLife.38215.009

**Figure supplement 1.** Activation of muscles with hip extension actions over the adaptation period.

DOI: https://doi.org/10.7554/eLife.38215.007

**Figure supplement 1—source data 1.** Integrated activity of hip extensor muscles during stance phase of locomotion.

DOI: https://doi.org/10.7554/eLife.38215.008

The effects of increased RF activity on joint kinematics should be most apparent at the end of the stance phase since quadriceps muscle activations are largest in late stance (*Figures 2* and *3*). As illustrated in *Figure 6a–b*, the hip angle was significantly decreased after the first week (i.e. more flexed; $p_{week1}$ = 1, $p_{week2}$ = 0.003, $p_{week7}$ = 0.032), and the ankle angle was significantly increased at all time points (i.e. more extended; $p_{week1}$ = 0.030, $p_{week2}$ = 0.004, $p_{week7}$ = 0.038) as compared to baseline. On the other hand, although the knee angle was significantly reduced in the first two weeks after VL paralysis ($p_{week1}$ <0.001, $p_{week2}$ = 0.031), it was not significantly different from baseline at 7 weeks ($p_{week7}$ = 0.167). We found equivalent results for joint angles measured earlier in the stance phase (*Figure 6—figure supplement 1*), at another point within the quadriceps burst. These results show that increased activation of RF, although compensating for the knee extension lost after

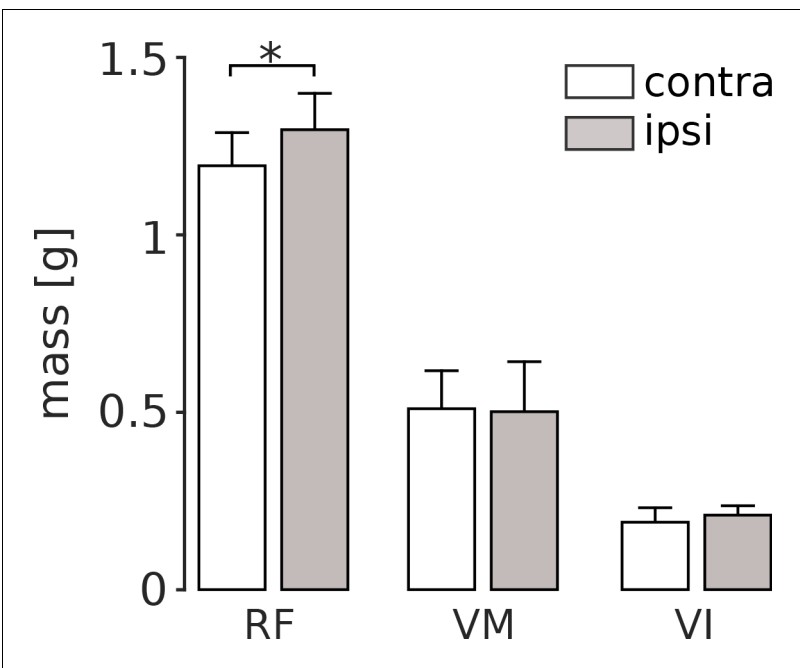

**Figure 4.** Quadriceps muscle masses measured 8 weeks after VL paralysis. At the end of the experiment, we measured the mass of RF, VM and VI both in the hindlimb affected by VL denervation (ipsi), and in the contralateral limb (contra). There was a significant increase in the mass of the ipsilateral RF (p<0.05), and no significant difference in the mass of VM or VI. Bars are averages ± s.d. across animals; N = 6 for each bar.
DOI: https://doi.org/10.7554/eLife.38215.010

The following source data is available for figure 4:

**Source data 1.** Quadriceps muscle masses in the ipsilateral and the contralateral hindlimbs.
DOI: https://doi.org/10.7554/eLife.38215.011

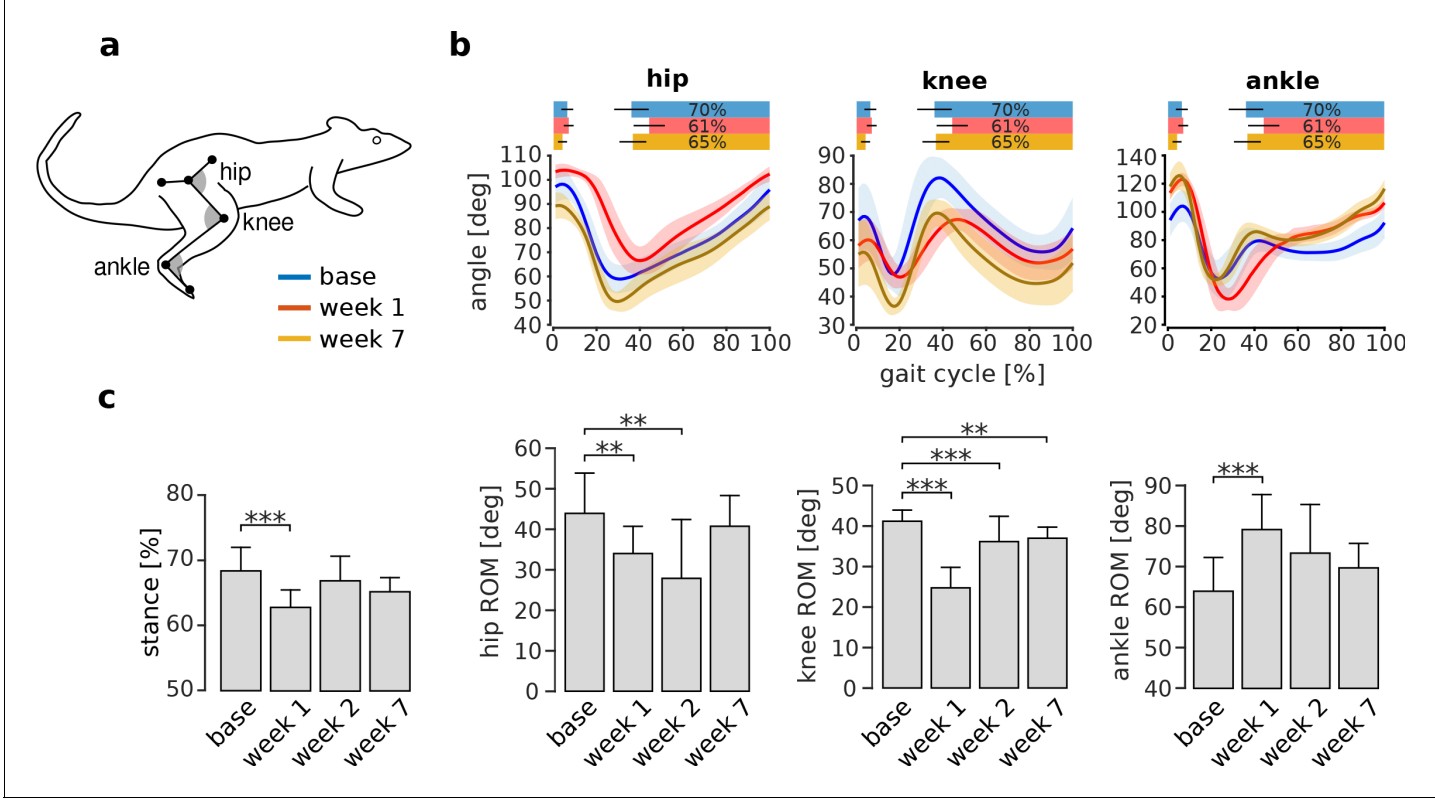

**Figure 5.** Adaptation of overall kinematics following VL paralysis. Panel (**a**) illustrates the joint angle conventions used for these analyses. For each angle, smaller numbers indicate greater flexion (i.e. the gray angles in the figure are smaller for flexed angles). Panel (**b**) shows the joint angle trajectories at the hip, knee, and ankle during baseline (blue), week 1 (red), and week 7 (orange) after VL paralysis for one animal (hip and knee: Ns = 72, 31 and 50 strides for base, week 1 and week 7; ankle: Ns = 49, 13, 36). The bars at the top of each plot indicate the periods of stance phase for each time point of adaptation. Panel (**c**) reports the percentage of stance and the range of motion for each joint angle before and after VL paralysis for all animals (averages ± s.d. across animals). Stance percentage: N = 6, 4, 5, 5 animals for base, week 1, week 2 and week 7, and an average number of strides of Ns = 48, 41, 34, 59 respectively. Hip ROM: N = 5, 4, 4, 4 animals, and Ns = 58, 47, 50, 93 strides. Knee ROM: N = 6, 5, 4, 5 animals, and Ns = 64, 57, 50, 86 strides. Ankle ROM: N = 6, 4, 4, 5 animals and Ns = 59, 55, 44, 82 strides. **p<0.01; ***p<0.001.
DOI: https://doi.org/10.7554/eLife.38215.012
The following source data is available for figure 5:

**Source data 1.** Overall kinematic features before and after VL devervation.
DOI: https://doi.org/10.7554/eLife.38215.013

VL paralysis, introduced persistent deviations at other joints. In particular, the increased hip flexion is consistent with the extra hip flexor torque produced by RF.

## Restoration of global limb kinematics following VL paralysis

Although the results of *Figure 6a–b* demonstrate persistent changes in local joint kinematics, the CNS might restore global features of limb kinematics such as overall limb length or limb angle. As illustrated in *Figure 6c*, both limb length and limb angle measured at the end of stance were initially altered after VL paralysis (limb length: $p_{week1}$ <0.001; limb angle: $p_{week1}$ = 0.002). However, these alterations disappeared over the adaptation period so that these measures were not significantly different from baseline levels (limb length: $p_{week2}$ = 0.142, $p_{week7}$ = 1; limb angle: $p_{week2}$ = 0.141, $p_{week7}$ = 1). We found equivalent results for global kinematics measured earlier in the stance phase (*Figure 6—figure supplement 1*). These results suggest that the CNS adopts a control strategy that limits internal joint stresses and restores global aspects of locomotion.

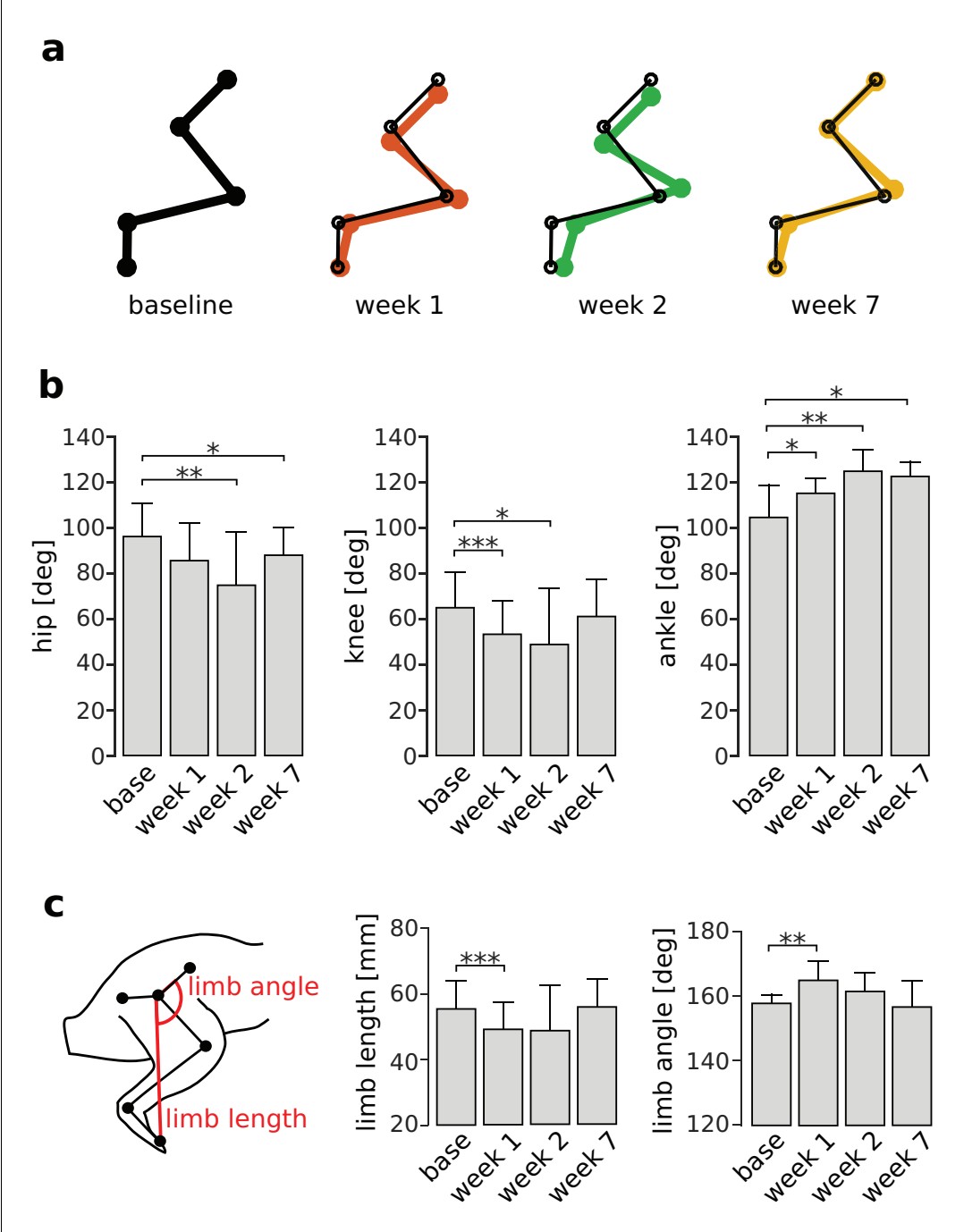

**Figure 6.** Persistent deviations in hindlimb kinematics following VL paralysis. Panel (**a**) illustrates limb configurations at the end of the stance phase at baseline (black), week 1 (red), week 2 (green), and week 7 (orange) after VL paralysis. For ease of comparison, thin black lines indicating the baseline limb configuration are overlaid on the limb configurations shown for weeks 1, 2, 7. Joint angles were averaged across animals and used to plot the configurations shown in the figure. Panel (**b**) shows the angles at the hip, knee, and ankle averaged across animals before and after VL paralysis (averages ± s.d. across animals). Hip: N = 5, 4, 4, 5 animals for base, week 1, week 2 and week 7, and an average number of strides of $N_s$ = 54, 47, 44, 74 strides. Knee: N = 6, 5, 5, 5 animals, and $N_s$ = 58, 51, 36, 74 strides. Ankle: N = 6, 4, 5, 5 animals, and $N_s$ = 54, 51, 34, 73 strides. Panel (**c**) shows the global kinematic parameters limb angle and limb length averaged across animals before and after VL paralysis (averages ± s.d. across animals). Limb length: N = 6, 4, 5, 5 animals, and $N_s$ = 59, 52, 34, 74 strides. Limb angle: N = 5, 3, 4, 5 animals, and $N_s$ = 54, 48, 42, 74 strides. *$p<0.05$; **$p<0.01$; ***$p<0.001$.

DOI: https://doi.org/10.7554/eLife.38215.014

The following source data and figure supplements are available for figure 6:

*Figure 6 continued on next page*

*Figure 6 continued*

**Source data 1.** Joint kinematics and global limb kinematics at the end of the stance phase of locomotion.

DOI: https://doi.org/10.7554/eLife.38215.017

**Figure supplement 1.** Joint configurations measured at mid-stance.

DOI: https://doi.org/10.7554/eLife.38215.015

**Figure supplement 1—source data 1.** Joint kinematics and global limb kinematics in the middle of the stance phase of locomotion.

DOI: https://doi.org/10.7554/eLife.38215.016

## Discussion

We hypothesized that the CNS activates muscles to minimize internal joint stresses while achieving task requirements. We found that animals compensated for VL paralysis by increasing activation of RF while maintaining the activation of VM at levels similar to those observed before the perturbation. This strategy restores the knee extension torque lost after VL paralysis while causing minimal additional mediolateral patellar forces. The increased RF activation resulted in a persistent perturbation in joint kinematics due to the extra hip flexor torque from RF, but global aspects of limb kinematics were restored. These results suggest that the CNS considers internal joint stresses and strains when choosing muscle activations.

The main result of this study is the observation that animals compensated for VL paralysis by preferentially increasing activation of RF. If one only considers muscle contributions to task performance, this adaptation strategy is counterintuitive; because VM and VL have redundant contributions to knee extension movements (*Sandercock et al., 2018*), restoring task performance would be most easily accomplished by increasing activation of VM (*Figure 1b*). This strategy would restore the same joint kinematics observed before VL paralysis without requiring any additional compensation. In contrast, increased activation of RF, while restoring the knee extension torque lost by VL paralysis, also introduces an additional flexion torque at the hip (*Johnson et al., 2008*; *Greene, 1963*) (*Figure 1*). Thus, this strategy would either require additional changes in other muscles across the limb, or introduce residual deviations in joint kinematics. The increased activation of RF we observed is therefore difficult to reconcile with strategies for muscle coordination focused solely on considerations of task performance.

This strategy is consistent, however, with the hypothesis that the CNS takes into account the effect of muscle activity on the stresses and strains within a joint. Increasing activation of RF restores knee extension function while producing negligible additional mediolateral forces on the patella (*Figure 1*). On the other hand, higher activation of VM would increase the medial force on the patella and cause excess patellofemoral loading. Hence, our results suggest that the CNS considers these mediolateral forces when selecting muscle activations to compensate for VL paralysis.

We also observed a selective increase in the mass of RF that likely reflects a use-dependent strengthening of that muscle, although we did not confirm this by measuring peak forces or myosin content (*Blaauw et al., 2013*). This result indirectly confirms the preferential increase of RF activity after VL paralysis. Furthermore, it suggests that at late time points of the adaptation, the loss of VL was compensated both by higher neural activity and by additional force generating capability of RF. Finally, this result suggests that the increased RF activation was not specific to treadmill locomotion, but was also used in behaviors we did not measure here; if there was increased VM or VI activation in other behaviors, we would have observed an increase in the mass of those muscles. The observation of no change in the mass of VI also suggests that there was no consistent change in the activation of this muscle during the adaptation period, though we did not record from VI in these experiments due its small size and inaccessibility for implantation.

We did not observe a complete silencing of VM, even after 7 weeks of adaptation: VM activation levels were on average similar to those observed before VL paralysis. This implies that a residual medial force on the patella persisted throughout the adaptation period. Such residual VM activity might reflect a compromise between unbalanced mediolateral patellar forces and perturbations in joint kinematics caused by the increased activation of RF. In the context of optimization approaches to motor control (*Todorov and Jordan, 2002*; *Scott, 2004*), this result suggests a cost function that includes terms for task performance as well as internal joint stresses. Alternatively, the residual activation of VM might reflect habitual persistence of the strategy used by the CNS prior to VL paralysis

(*de Rugy et al., 2012*; *Loeb, 2012*). Although our study cannot distinguish between these two explanations, in either case the fact that RF and not VM activity was increased demonstrates the influence of criteria related to the state of internal joint structures.

The particular aspects of task performance that drive adaptation are potentially complex. Although muscle paralysis caused persistent changes at individual joint kinematics, we observed restoration of global limb kinematics. This result is consistent with ideas such as those used in uncontrolled manifold analyses (*Latash et al., 2007*) and with previous results examining adaptation following paralysis of triceps surae in rats and cats (*Chang et al., 2009*; *Bauman and Chang, 2013*). In this context, our results suggest a hierarchical control of movement, in which higher level variables related to task goals (i.e. global limb variables) are preserved after VL paralysis while lower level variables related to task execution (i.e. individual joint angles) are allowed to vary.

Many criteria have been proposed to influence the activation of muscles during behavior (*Alessandro, 2016*; *Prilutsky, 2000*), but they generally do not consider the internal state of joints. Although some of these criteria might indirectly minimize joint stresses and strains (e.g. minimum jerk [*Flash and Hogan, 1985*] or torque change [*Uno et al., 1989*]), they would not predict the selective increase in RF activation observed here. Including criteria that explicitly consider the consequences of muscle activation on joint stresses and strains might improve the predictions of such optimization approaches.

Alternatively, minimization of internal joint stresses and strains in intact subjects might be accomplished without explicit control. The co-evolution of neural control and musculoskeletal anatomy (*Valero-Cuevas, 2015*) might result in a neuromechanical system such that muscle activations optimizing task performance criteria also optimize internal joint state criteria. For example, when VM and VL are intact, a criterion that minimizes muscle metabolism (*Kistemaker et al., 2014*; *Prilutsky, 2000*), or that considers signal dependent variability (*Harris and Wolpert, 1998*; *Haruno and Wolpert, 2005*) would predict a distributed activation of VM and VL. This distributed activation of VM and VL might result from optimization over evolutionary time scales (*Giszter, 2015*) and be implemented in coordinative structures such as muscle synergies (*Tresch and Jarc, 2009*; *Alessandro et al., 2013a*; *Alessandro et al., 2013b*) within the spinal cord (*Levine et al., 2014*; *Hart and Giszter, 2010*; *Takei et al., 2017*) or other brain regions (*Overduin et al., 2015*). This distributed activation could in turn balance net mediolateral patellar forces, even though such a balance was not explicitly considered. After injuries that alter limb properties (*Gutierrez et al., 2009*; *O'Connor et al., 1993*), this implicit regulation of internal joint state might be compromised (*Needle et al., 2014*). In these situations, the CNS might be forced into a solution that requires a tradeoff between task performance and internal joint stresses and strains, such as that observed in our experiments.

These results have implications to clinical syndromes affecting joint health, such as knee osteoarthritis (*Felson, 2013*; *Mahmoudian et al., 2017*) and patellofemoral pain (*Smith et al., 2018*) (PFP) in humans. Although many factors have been shown to contribute to PFP (*Smith et al., 2018*; *Fagan and Delahunt, 2008*), one common suggestion is that PFP results from an imbalance in mediolateral patellar forces (*Pal et al., 2012*). Our results show that although the CNS regulates these mediolateral forces, this regulation is imperfect since VM activation persisted after VL paralysis. PFP might result from similarly imperfect regulation of patellar forces, potentially reflecting a compromise between criteria related to task performance and internal joint states. More broadly, our experiments highlight the importance of considering the consequences of rehabilitation on internal joint stresses and strains following injuries (*Farrokhi et al., 2013*; *Shull et al., 2013*; *Simic et al., 2011*), rather than solely focusing on restoration of task performance (*Hollands et al., 2012*).

Regulation of internal joint stresses and strains, either in intact subjects or following injury, might be accomplished by feedforward or feedback mechanisms. Sensory receptors in the joint can convey information about strains in ligaments or stresses between bones. Activity in these receptors can be conveyed to spinal interneuronal populations (*Solomonow et al., 1987*; *Iles et al., 1990*) and other areas (*Sjölander et al., 2002*) to drive rapid feedback control of muscles (*Ferrell, 1980*) or longer term adaptation of muscle activations in order to avoid potential injury. The CNS could also derive information about internal joint structures indirectly from muscle proprioceptors (*Wilmink and Nichols, 2003*), combining information about muscle forces and lengths with 'internal' models of muscle anatomy to estimate joint stresses and strains. Similarly, predictive models of muscle actions, combined with efference copy of motor commands, could be used to make feedforward predictions of

internal joint stresses and strains. The strong coherence between VM and VL across frequencies (*Laine et al., 2015*) observed in humans is consistent with the importance of coordinated activation of these muscles (*Brøchner Nielsen et al., 2017*), whether by feedforward or feedback processes. This coordinated muscle activation is also similar to ideas of muscle synergies (*Tresch and Jarc, 2009*; *Alessandro et al., 2013a*; *Alessandro et al., 2013b*), so that the balanced activation of muscles within a synergy would reflect regulation of joint stresses and strains. Note that neural control of internal joint states might be relatively coarse, only responding to large deviations in joint states inducing discomfort or pain (*Grigg and Greenspan, 1977*). Such a coarse control of joint state might reflect the 'good enough' habitual strategy (*de Rugy et al., 2012*) described above, potentially resulting in the residual activation of VM observed in our experiments.

Further experiments will be necessary to better understand the neural control of joint stresses and strains. Although the muscle paralysis by denervation used in these experiments has been used in many studies to examine neural adaptation strategies (*Bauman and Chang, 2013*; *Dambreville et al., 2016*; *Frigon and Rossignol, 2007*; *Bennett et al., 2012*; *Bouyer et al., 2001*; *Maas et al., 2007*), the loss of sensory information from the paralyzed muscle might contribute to the observed changes in muscle activation (*Akay et al., 2014*). Perturbations directly manipulating patellar forces or paralyzing VM or RF might provide complementary experiments to further evaluate this hypothesis. Cutting the nerves to VM and RF selectively, however, is difficult because they branch extensively immediately after leaving the common quadriceps nerve, making them hard to isolate (*Greene, 1963*). The nerve to VL was simpler to cut because it persists as a single trunk for a long distance from the common quadriceps nerve.

Our experiments exploited the unique mechanical properties of the rat knee joint: the clear distinction between muscle contributions to task performance and to patellar forces in the rat allowed us to make simple predictions about adaptation strategies consistent or inconsistent with neural regulation of internal joint stresses and strains. Such distinctions are more difficult to make in humans because the patellar mobility at extended knee angles complicates how quadriceps muscles affect both patellar mechanics and limb kinematics (*Farahmand et al., 1998*). The rat knee joint might therefore provide a tractable experimental model to examine the neural control of joint stresses and strains in both healthy subjects and following injury.

## Materials and methods

### Ethical approval

We performed experiments on female Sprague Dawley rats (n = 6). All the procedures were approved by the Animal Care Committee of Northwestern University.

### Experimental protocol

We first trained rats to maintain a stable gait pattern during treadmill locomotion. We then implanted chronic EMG electrodes in multiple hindlimb muscles and allowed animals to recover for at least 10 days following implantation. We recorded kinematic and EMG activity during treadmill locomotion before (baseline) and at different times after VL paralysis: within one week after the procedure (week 1), within the second week (week 2), and within the seventh week (week 7). At the end of data collection, animals were euthanized, muscle masses weighed, and electrode location verified. The EMG channels corresponding to misplaced electrodes were not included in the analysis.

### Implantation of EMG electrodes

We anesthetized animals with isoflurane (3% in O2 ~ 2 l/min), and prepared them for aseptic surgery. We implanted pairs of electrodes in seven muscles: vastus lateralis (VL), vastus medialis (VM), rectus femoris (RF), semimembranosus (SM), biceps femoris posterior (BFp), tibialis anterior (TA), and caudal gracilis (GRc). Knots placed on either side of the muscle secured the exposed electrode sites within the muscle belly. The electrode leads were tunneled subcutaneously to a connector on the back of the animal. These methods have been described in more details previously (*Tysseling et al., 2013*).

## Paralysis of VL

During aseptic surgery (isoflurane, 3% in O2 ~ 2 l/min), we exposed the quadriceps nerve plexus, and isolated the branch to VL. We anesthetized this branch with lidocaine, tied it off in two locations with silk suture to prevent re-innervation, and then cut between the sutures.

## EMGs and kinematics during locomotion

Before each recording session, we applied retro-reflective markers on the shaved skin (see *Data acquisition and processing*) under brief isoflurane anesthesia (2–3% in O2 ~ 2 l/min). Markers were placed on bony landmarks on the hindlimb skin (rostral and caudal tips of the pelvis, hip, knee, ankle and first digit, see *Figure 5a*). The EMG connector was attached via cable to the amplifier, and the animal placed on the treadmill. We waited at least 30 min after anesthesia before collecting behavioral data. Each recording session consisted of two minutes of locomotion at the maximum comfortable speed (12–15 m/min) and incline (usually +25%) to induce strong quadriceps activations. One animal was unable to walk at an incline of +25%, and so for that animal we recorded data from level treadmill walking.

## Data acquisition and processing

The 3D position of markers was tracked using a motion capture system (Vicon) at a frequency of 200 Hz. These signals were low-pass filtered offline at a cut-off frequency of 10 Hz (5th order Butterworth). In order to reduce errors due to differential movements of the skin (*Filipe et al., 2006*; *Bauman and Chang, 2010*), we estimated the 3D position of the knee by triangulation, using the lengths of the femur and the tibia (*Bauman and Chang, 2010*). Differential EMG signals were amplified (1000X), band-pass filtered (30–1000 Hz) and notch filtered (60 Hz), and then digitalized (5000 Hz). The digitalized signals were further band-pass filtered offline to remove motion artifacts (5–500 Hz, 4th order Butterworth), rectified, and envelopes were computed from the rectified signals.

We segmented both kinematic and EMG envelopes into separate strides, defining the beginning of each stride as the onset of activation of TA during bouts of stable walking. To obtain a consistent dataset that reflected the behavioral condition of the experiment, we only considered strides with durations within 1.5 standard deviations from the mean. We also excluded strides with clear EMG artifacts that could occur when the cable hit the side of the treadmill, as identified using Tukey outlier analysis (i.e. identifying EMG values that were 1.5 interquartiles above the upper quartile of the maxima across steps). Further, we discarded strides that had more than 20% of missing values in the kinematic signals due to malfunctioning of the acquisition system or due to occlusions of the retro-reflective markers. Application of these inclusion criteria resulted in data sets with, on average, 65 strides for each recording session. Strides were then time-normalized to 100 samples for further analysis.

We characterized each stride by means of several kinematic and EMG features. For kinematic analysis, we calculated the percentage of stance (i.e. duration of the stance phase divided by the duration of the entire stride, with stance phase defined as the interval between foot-strike and foot-off). We calculated local joint angles (hip, knee, and ankle) as illustrated in *Figure 5a*. Furthermore, we calculated more global features describing overall limb kinematics as illustrated in *Figure 6c*: the length of the global limb vector (defined as the vector from the hip to the toe), and the limb angle (defined as the angle between the limb vector and the vector from the hip to the rostral pelvis). Finally, we calculated the range of motion of each joint (ROM, i.e. the difference between maximum and minimum joint angles for each stride). All angles were calculated using the projections of the markers onto the sagittal plane of the treadmill, and were defined such that zero degrees reflected complete joint flexion and increasing angles reflected increasing amounts of extension. For the EMG analysis, we computed the integral of the EMG envelopes' extension burst (i.e. muscle activity related to the preparation and the execution of the stance phase), which represents the muscle activity contributing to limb extension. We normalized this measure to the pre-paralysis mean integrated activity for each muscle and subject, obtaining the relative change of activation with respect to this baseline.

## Statistical analyses

We employed Linear Mixed Effect Models (LMEM) to analyze both kinematic and EMG data using the nlme package (*Pinheiro et al., 2017*) in the R environment. The use of LMEMs allows us to exploit the fact that we have large numbers of observations for each animal across the adaptation period, obtaining the maximal statistical power from a relatively small number of animals. LMEMs also allow us to analyze samples of different size (e.g. unequal numbers of strides across days and animals), to cope with missing data (potentially due to the inclusion criteria described above), and to take into account variability at different levels of the dataset (e.g. across strides, across days, and across subjects). To confirm that our dataset met the assumption of Gaussian distribution and independence of residuals and random effects (*Pinheiro and Bates, 2000*), we visually inspected the distributions using qq-plots and histograms. After fitting the LMEMs, we tested our specific hypotheses of interest by performing post-hoc tests on the model parameters and using Bonferroni corrections to adjust the obtained p-values. Note that because of this correction, the adjusted p-values can be equal to 1. We considered tests to be statistically significant if their p-values were lower than the 0.05 significance level.

For the kinematic analysis, we fit a LMEM for each measure, using the value of the feature as the dependent variable and the timing after VL paralysis as the independent variable. In order to take inter-subject variability into account, we considered *subject* as a random effect on both slope and intercept (*Pinheiro and Bates, 2000*). This method is essentially a more powerful version of a repeated measures ANOVA, and it has been proven effective at estimating degrees of freedom and standard errors, hence enabling accurate statistical tests (*Barr et al., 2013*). We then performed post-hoc tests to evaluate whether the values of the features at a given week were different from their baseline values.

To analyze muscle activity, we initially log-transformed the data to reduce the skewness of EMG features. This rendered the distribution of the dataset more symmetrical and the distribution of residuals approximately Gaussian. For the comparison between VM and RF, we fit a LMEM with normalized integrated EMG as the dependent variable, and the timing after VL paralysis as well as muscle identity (i.e. whether the muscle was VM or RF) as independent variables, including in the model also an interaction term between these independent variables. We considered *subject* as a random effect on both intercept and slope, and we nested the variability across strides within each subject and recording session. We performed post-hoc tests to evaluate: (1) whether the normalized activity in RF at any week after VL paralysis was greater than the activity in VM; and (2) whether the normalized activity of each muscle at a given week was different from its baseline value. For the analysis of the other muscles (*Figure 3—figure supplement 1*), we fit a LMEM for each signal using the normalized integrated EMG as the dependent variable, the timing after VL paralysis as the independent variable, and considering *subject* as a random effect on both slope and intercept. Then, we used post-hoc tests to evaluate whether the normalized activity at a given week was different from its baseline value.

To compare the masses of the quadriceps in the denervated limb to those in the intact limb, we performed paired t-tests for each muscle, and we adjusted the obtained p-values with Bonferroni corrections.

## Acknowledgments

This research was supported by the NIH grant number NS086973 (MCT).

## Additional information

### Funding

| Funder | Grant reference number | Author |
|---|---|---|
| National Institutes of Health | NS086973 | Matthew Tresch |

The funders had no role in study design, data collection and interpretation, or the decision to submit the work for publication.

## Author contributions

Cristiano Alessandro, Conceptualization, Data curation, Software, Formal analysis, Validation, Investigation, Visualization, Methodology, Writing—original draft, Writing—review and editing; Benjamin A Rellinger, Conceptualization, Investigation; Filipe Oliveira Barroso, Formal analysis, Methodology; Matthew C Tresch, Conceptualization, Resources, Supervision, Funding acquisition, Validation, Methodology, Writing—original draft, Project administration, Writing—review and editing

## Author ORCIDs

Cristiano Alessandro (iD) https://orcid.org/0000-0003-0655-4189
Filipe Oliveira Barroso (iD) http://orcid.org/0000-0003-0228-6447
Matthew C Tresch (iD) http://orcid.org/0000-0001-9994-3989

## Ethics

Animal experimentation: All of the animals were handled according to approved institutional animal care and use committee (IACUC) protocols (#IS00000628) of the Northwestern University. The protocol was approved by the Animal Care Committee of Northwestern University.

## Decision letter and Author response

Decision letter https://doi.org/10.7554/eLife.38215.020
Author response https://doi.org/10.7554/eLife.38215.021

# Additional files

## Supplementary files

• Transparent reporting form
DOI: https://doi.org/10.7554/eLife.38215.018

## Data availability

We have uploaded files with all source data and code for data analysis for each of the figures showing primary data.

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
