## [Decision Letter]

Thank you for submitting your article "Adaptation after vastus lateralis denervation in rats demonstrates neural regulation of joint stresses and strains" for consideration by *eLife*. Your article has been reviewed by three peer reviewers, including Richard Nichols as the Reviewing Editor and Reviewer #1, and the evaluation has been overseen by Richard Ivry as the Senior Editor. The following individuals involved in review of your submission have agreed to reveal their identity: Michel A Lemay (Reviewer #2) and Simon Giszter (Reviewer #3).

The reviewers have discussed the reviews with one another and the Reviewing Editor has drafted this decision to help you prepare a revised submission.

Summary:

In this novel and important study, evidence was obtained that altered muscular recruitment patterns following denervation of one of a group of synergistic muscles reflect a balance between meeting kinematic requirements and minimizing stress on the articular surfaces. The findings are significant in revealing previously underappreciated factors that determine how muscles in redundant groups are recruited. The issue of muscle selection in redundant systems has been considered of major significance in motor control research, and the results of this study provide an important hypothesis in this regard.

Essential revisions:

1) It was suggested that the authors might amplify the Discussion by referring to sensory pathways from the joint capsule that could potentially mediate the adaptation to the unbalanced forces within the joint.

2) An interesting point worthy of discussion is that the mechanisms underlying the observed adaptation might be predictive, since the damage to the joint would develop over time after the injury.

3) Consider addressing the idea of Young-Hui Chang that stabilization of task-level variables (e.g. whole limb length) might be relevant in the selection process due to the multi-joint actions of some muscles.

4) Figure 3B, subsection “Preferential increase in RF activation following VL paralysis”: Please clarify whether the EMG was integrated across the whole step cycle, since the activity of RF exhibited two peaks, one in each phase.

5) Figure 6: It appears that total limb length observed in controls was preserved following the denervation, at least after 7 weeks. Was this a general finding (see point 3)? Feedback of limb length might be part of the mechanism of adaptation.

6) Subsection “Data acquisition and processing”, first paragraph: Note that Bauman and Chang (2010) made a direct comparison of kinematics using skin markers and x-ray images to show the inaccuracies resulting from the use of skin markers.

---

## [Author Response]

Essential revisions:1) It was suggested that the authors might amplify the Discussion by referring to sensory pathways from the joint capsule that could potentially mediate the adaptation to the unbalanced forces within the joint.

We have revised the Discussion to include these points, elaborating on the role of joint receptors in mediating control of internal joint variables (Discussion, tenth paragraph).

2) An interesting point worthy of discussion is that the mechanisms underlying the observed adaptation might be predictive, since the damage to the joint would develop over time after the injury.

This is an interesting point and we have included it in the Discussion (tenth paragraph).

3) Consider addressing the idea of Young-Hui Chang that stabilization of task-level variables (e.g. whole limb length) might be relevant in the selection process due to the multi-joint actions of some muscles.

This is a good observation. We had noticed this as well and were preparing a separate publication on this point, but at the reviewers’ suggestion have included the analysis here. We analyzed the changes in limb length and limb angle over the adaptation period (Figure 6C). As suggested by the reviewer, we found that although there were changes in these global parameters in the first week following VL paralysis, these parameters returned to baseline, pre-paralysis values by the end of the adaptation period. We have included these analysis in the Materials and methods (subsection “Data acquisition and processing” third paragraph), Results (subsection “Restoration of global limb kinematics following VL paralysis”), and briefly consider their implications in the Discussion (second paragraph). We also added these analyses of global kinematics to Figure 6—figure supplement 1 illustrating kinematic adaptation at mid-stance. We have made edits through the paper to reflect the distinction between ‘local’ joint angles and ‘global’ limb angles.

4) Figure 3B, subsection “Preferential increase in RF activation following VL paralysis”: Please clarify whether the EMG was integrated across the whole step cycle, since the activity of RF exhibited two peaks, one in each phase.

The activity of VM and RF was only integrated over the extension burst. We have clarified this point in the Results (subsection “Preferential increase in RF activation following VL paralysis”) and in the Figure 3 legend.

5) Figure 6: It appears that total limb length observed in controls was preserved following the denervation, at least after 7 weeks. Was this a general finding (see point 3)? Feedback of limb length might be part of the mechanism of adaptation.

See response to point 3 above.

6) Subsection “Data acquisition and processing”, first paragraph: Note that Bauman and Chang (2010) made a direct comparison of kinematics using skin markers and x-ray images to show the inaccuracies resulting from the use of skin markers.

We have included this reference as requested. We had cited this work in the previous manuscript for our estimation of knee position, but now cite it for that point as well as for the point about skin marker inaccuracies (subsection “Data acquisition and processing”).